# Synthesis of Calamitic Fluorinated Mesogens with Complex Crystallization Behavior

**DOI:** 10.3390/molecules28248002

**Published:** 2023-12-08

**Authors:** Denis Anokhin, Alina Maryasevskaya, Ainur Abukaev, Umut Ugur Ozkose, Alexander Buglakov, Dimitri A. Ivanov, Bruno Améduri

**Affiliations:** 1Faculty of Chemistry, Lomonosov Moscow State University, GSP-1, 1-3 Leninskiye Gory, 119991 Moscow, Russiaainurabukaev@gmail.com (A.A.); buglakov@polly.phys.msu.ru (A.B.); dimitri.ivanov@uha.fr (D.A.I.); 2Scientific Center for Genetics and Life Sciences, Sirius University of Science and Technology, 1 Olympic Ave., 354340 Sochi, Russia; 3Institut Charles Gerhardt, CNRS, University of Montpellier, Ecole Nationale Supérieure de Chimie de Montpellier, 34000 Montpellier, France; umut-ugur.ozkose@umontpellier.fr; 4Department of Chemistry, Faculty of Science and Letters, Piri Reis University, Tuzla, Istanbul 34940, Turkey; 5A. N. Nesmeyanov Institute of Organoelement Compounds RAS, Vavilova ul., 28, 119334 Moscow, Russia; 6Institut de Sciences des Matériaux de Mulhouse-IS2M, CNRS UMR 7361, Jean Starcky 15, 68057 Mulhouse, France

**Keywords:** DSC, fluorinated compounds, liquid crystals, melting point, smectic

## Abstract

This work presents the synthesis and self-organization of the calamitic fluorinated mesogen, 1,1,2,2–tetrafluoro–2–(1,1,2,2–tetrafluoro–4–iodobutoxy)ethanesulfonic acid, a potential model for perfluorosulfonic acid membranes (PFSA). The compound is derived in three steps from 1,1,2,2–tetrafluoro–2–(1,1,2,2–tetrafluoro–2–iodoethoxy)ethanesulfonyl fluoride, achieving a 78% overall yield. The resulting compound exhibits intricate thermal behavior. At 150 °C, a crystal-to-crystal transition is observed due to the partial disordering of calamitic molecules, which is followed by isotropization at 218 °C. Upon cooling, sample ordering occurs through the formation of large smectic liquid crystalline phase domains. This thermotropic state transforms into a layered crystal phase at lower temperatures, characterized by alternating hydrophilic and hydrophobic layers. Using X-ray diffraction, crystalline unit cell models at both room temperature and 170 °C were proposed. Computer simulations of the molecule across varying temperatures support the idea that thermal transitions correlate with a loss of molecular orientation. Importantly, the study underscores the pivotal role of precursor self-organization in aligning channels during membrane fabrication, ensuring controlled and oriented positioning.

## 1. Introduction

In recent years, increasing concern has been raised about the growing environmental burden caused by the use of fossil fuels and the overall increase in energy consumption, especially in EU countries [1,2,3]. Searching for clean energy, particularly decarbonized sources, is relevant, and according to various studies, photovoltaics (PVs) [4,5], lithium-ion batteries (LIBs) [6], windmills [7], and fuel cells [8,9] appear to be innovative and useful means for overcoming such challenges. The fuel cell is a device that converts the electrochemical energy of a fuel’s oxidation–reduction into electricity, heat, and water. One of the most important components is the membrane, and among them, perfluorosulfonic acid ones (PFSA) such as Nafion^®^, Flemion^®^, Aquivion^®^, and 3M^®^ membranes are commercially available from Chemours, AGC, Solvay, and 3M Innovative, respectively [10]. Many articles have been published on the synthesis, characterization, processing, and electrochemical applications of portable, small, and bigger embedded devices (vehicles that are already produced by Toyota [11], Honda, or Hyundai). It has been found that the proton transport properties of PFSA membranes strongly depend on their morphology [12]. The self-assembly of fluorinated side-groups determines the volume fraction, topology, and size of water channels. Despite numerous publications on the correlation between chemical structure, preparation conditions, and final supramolecular morphology of polymeric membranes, this question remains open due to polymer specificity, such as slow structure formation and polydispersity of model ionomers [13,14,15]. To better understand the self-assembly process in ion-transporting membranes, it would be of interest to propose models (i.e., small molecules) capable of self-assembling under conditions similar to those required for fluorinated comb-like copolymers. Such semi-rigid amphiphilic compounds, known as calamitics, are known to form layered, cylindrical, or cubic superstructures as a function of molecular shape, temperature, humidity, etc. [16,17,18]. For example, Berrod et al. [19] suggested original perfluorosulfonic acid molecules that can reproduce the phase-separated morphology of PFSA ionomer membranes used as electrolytes in fuel cells.

Investigations of thermotropic mesogens are also of great interest from the fundamental point of view, as the peculiarities of the structure and assembly mechanisms of liquid crystals is yet to be fully understood [20,21,22]. Specific details on molecular level arrangements, conditions of transitions between different phases, and so on can be resolved through various simulation techniques [23,24], and comparisons between experimental and theoretical data can be used for the refinement of force fields and models, leading to a better understanding of the physical reasons for the mentioned assemblies. To date, there are numerous modeling studies related to thermotropic mesogens [23,24,25,26,27] which show the high ability of modern methods to predict and reveal some features of the structuring of mesogens qualitatively as well as quantitatively.

## 2. Results and Discussion

### 2.1. Ethylenation of the Iodofluorocompounds

1,1,2,2–tetrafluoro–2–(1,1,2,2–tetrafluoro–4–iodobutoxy)ethanesulfonic acid was synthesized in three steps from 1,1,2,2–tetrafluoro–2–(1,1,2,2–tetrafluoro–2–iodoethoxy)ethanesulfonyl fluoride (Figure 1). (i) First, quantitative ethylenation was initiated by *tert*–butyl cyclohexyl peroxydicarbonate at 60 °C (that generates only the monoadduct as in previous works [28]). The vanishing of the signal at –70 ppm (highlighted by ^19^F NMR spectrum, Appendix A) and the presence of two complex multiplets at 2.3 and 3.0 ppm in the ^1^H NMR spectrum (Appendix A) evidenced the formation of the ethylenated product (with the insertion of one ethylene unit only). (ii) The second step enabled the modification of the sulfonyl fluoride end group into SO_3_Li under mild basic conditions to avoid dehydroiodination. (iii) The final step dealt with the acidification of the latter into sulfonic acid. These reactions were monitored using ^19^F, ^1^H, and IR spectroscopy (see Appendix A). The O–F frequencies (bending modes) centered at 1470 and 810 cm^−1^ (noted in the spectrum of the product bearing the SO_3_Li end-group) were absent after the hydrolysis reaction while the –S(O)(O)OH stretching vibrations of the sulfonic acid groups at 3377 cm^−1^ and 1026 cm^−1^ were present (Appendix A), indicating a successful hydrolysis reaction. Additionally, the hydrolysis of the –SO_2_F groups was confirmed by the ^19^F NMR spectrum that does not display any signal at +45 ppm, which is characteristic of the sulfonyl fluoride group [29]. The absence of this peak indicates that the hydrolysis reaction yield was quasi-quantitative.

The overall yield was 78% from 1,1,2,2–tetrafluoro–2–(1,1,2,2–tetrafluoro–2–iodoethoxy)ethanesulfonyl fluoride.

### 2.2. Structural Characteristics of I(CH_2_)_2_(CF_2_)_2_O(CF_2_)_2_SO_2_OH

The first DSC heating thermogram (Figure 1) exhibits a double endothermic peak within the temperature range of 80 to 110 °C, which may be attributed to the dehydration or dehydroiodination of the sample. The peak at 150 °C is likely due to a crystal-to-crystal transition, typical for amphiphilic calamitic molecules [30]. An intense peak at 218 °C corresponds to the sample’s isotropization [31]. The second heating curve shows only isotropization, indicating the irreversibility of the solid-to-solid transition.

Polarized optical microscopy of thin films of I(CH_2_)_2_(CF_2_)_2_O(CF_2_)_2_SO_2_OH at 216 °C reveals crystalline spherulites and rod-like domains of the smectic liquid crystalline (LC) phase (Figure 2a). It is plausible that the LC phase is thermotropic, which signifies that it is in equilibrium in a certain temperature range. During cooling, the nucleation rate increases, leading to the emergence of a grainy polycrystalline texture at room temperature (Figure 2b).

Structural analysis was performed using temperature-resolved small- (SAXS) and wide-angle (WAXS) X-ray scattering. At room temperature, powder diffraction indicates a highly ordered crystalline structure with numerous peaks in the WAXS region (Figure 3). The SAXS region shows peaks typical for a layered structure. The complete table of the peaks is presented in Appendix A. According to SAXS, the longest inter-layer distance is 2.72 nm. The high-intensity peaks at *s* = 4.3 and 4.9 nm^−1^ are associated with aluminum foil used as a sample container.

The crystalline structure observed can be assigned to a monoclinic unit cell with lattice parameters: *a* = 27.5 Å, *b* = 5.2 Å, *c* = 5.0 Å, and angles *α* = 81°, *β* = 90°, and *γ* = 90°. Temperature-dependent SAXS and WAXS measurements are presented in Figure 4a and Figure 4b, respectively. During heating, distinct changes in both peak positions and intensities are noticeable in the SAXS and WAXS regions. Specifically, shifts observed at 90 °C and 150 °C are attributed to dehydration and a crystal-to-crystal transition, respectively. The isotropization process at 220 °C becomes evident with the fading of sharp crystalline peaks.

Indexation of the high-temperature phase indicates the formation of a triclinic unit cell with the following parameters: *a* = 25.6 Å, *b* = 5.12 Å, *c* = 4.96 Å, *α* = 95°, *β* = 97°, *γ* = 88°.

To gain insight into the arrangement of I(CH_2_)_2_(CF_2_)_2_O(CF_2_)_2_SO_2_OH at different temperatures, all-atom molecular dynamics simulations of a box sized 100 Å × 30 Å × 30 Å, filled with 288 molecules, were conducted at room temperature, followed by a heating procedure with a temperature increase up to 280 °C. A detailed description of the modeling methodology is provided in the Materials and Methods section.

Initially, after system equilibration at *T* = 25 °C, the molecules are elongated and retain crystal ordering. Approximation of the lattice parameters, obtained through identification of the minimal symmetric parallelepiped [32] and averaged over the system, yielded the following values: *a* = 27.2 Å, *b* = 5.0 Å, *c* = 5.0 Å, *α* = 81.1°, *β* = 89.6°, *γ* = 90.4°. These parameters correspond to a cell with monoclinic symmetry and are in good agreement with experimental data (Appendix A).

After equilibration at room temperature, the box of I(CH_2_)_2_(CF_2_)_2_O(CF_2_)_2_SO_2_OH was slowly heated in the NPT (isobaric–isothermal) ensemble from 25 °C to 280 °C. Within this temperature range, the system undergoes a series of transitions, analyzed using the nematic order parameter *S* [33], which indicates whether the molecules’ main axis is oriented in one direction (*S* = 1) or whether their orientations are distributed isotropically (*S* = 0). The presence of a layered structure was monitored through the calculation of the static structure factor *S*(*k/k**).

Figure 5 shows the dependency of the nematic order parameter *S* on temperature; the insets provide typical snapshots of the system under different states and the normalized static structure factor for the chosen *T*. The latter was plotted against *k/k**, where *k** is the location of the first maximum. For the layered structure, positions of consequent peaks *k*(1):*k*(2):*k*(3):*k*(4) should satisfy the relations 1:2:3:4 [34], indicating the periodicity of layers. This relation breaks when layered ordering is disrupted. The positions of these peaks are marked with vertical lines.

Overall, the dependency of the nematic order parameter on temperature is qualitatively similar to what is expected for ellipsoidal mesogens [35,36], with *S* close to 1 at low temperatures (crystal) and decreasing monotonically as *T* increases. Sharp changes in the order parameter *S* indicate transitions between different states (crystal, smectic, nematic, isotropic), with the most pronounced drop at the nematic–isotropic transition.

In the interval between 25 °C and ~170 °C, the molecules maintain a layered arrangement (peaks at *k/k** = 1, 2, 3, 4 at *T* = 150 °C, Figure 5 (upper right), Appendix A) and retain a high orientational ordering parameter, which decreases from *S* = 0.9 to *S* = 0.64 upon heating. At *T* ~170 °C, layers begin to overlap, leading to the destruction of the smectic phase; at *T* = 180 °C, only the first two peaks of *S*(*k/k**) remain (Figure 5 (bottom right), Appendix A). However, the system maintains high orientational order for I(CH_2_)_2_(CF_2_)_2_O(CF_2_)_2_SO_2_OH (*S* ~0.64). Thus, at *T* ~170 °C, the smectic-to-nematic transition can be observed. The model temperature for this transition to the nematic phase was found to lie within the interval between 170 and 245 °C, with a monotonic decrease of *S* upon heating from 0.64 to 0.42. A subsequent increase in *T* leads to a sharp drop in the nematic order parameter to 0.19 at *T* = 250 °C, with the nematic-to-isotropic transition temperature occurring approximately at *T* ~245 °C, as shown in Appendix A.

Therefore, within the all-atom molecular dynamics framework, two pronounced transition temperatures were identified: smectic-to-nematic at *T* ~170 °C and nematic-to-isotropic at *T* ~245 °C, both of which are in good agreement with experimental data (Figure 3). The fact that the values derived from simulations are slightly higher than those observed experimentally (i.e., 150 and 230 °C, respectively) can be attributed to a finite-size effect or to inaccuracies within the chosen force field.

## 3. Materials and Methods

### 3.1. Chemicals

1,1,2,2–tetrafluoro–2–(1,1,2,2–tetrafluoro–2–iodoethoxy)ethanesulfonyl fluoride was supplied from Apollo (New York, NY, USA) while ethylene was provided by Air Liquide, Paris, France; Perkadox 16S was purchased from Akzo Chemicals (Amsterdam, The Netherlands) and *tert*–butanol and Li_2_CO_3_ from Sigma Aldrich (St. Louis, MO, USA).

### 3.2. Synthesis

Ethylenation of 1,1,2,2–tetrafluoro–2–(1,1,2,2–tetrafluoro–2–iodoethoxy) ethanesulfonyl fluoride occurred in 50 mL Hastelloyl (HC276) autoclave equipped with inlet and outlet valves, a manometer, a rupture disk, a mechanical stirrer and a controller to check the speeding rate and the temperature. A solution composed of di(*tert*–butylcyclohexylperoxy dicarbonate (Perkadox 16S) (0.247 g, 0.63 mmol, 0.8% eq.), 28 mL of *tert*–butanol, and 16.82 g (39.5 mmol; 0.55% eq.) of 1,1,2,2–tetrafluoro–2–(1,1,2,2–tetrafluoro–2–iodoethoxy)ethanesulfonyl fluoride was degassed via N_2_ bubbling for 30 min. During this time, the autoclave was put under vacuum (40 × 10^−3^ bar) for 30 min to remove the residual traces of oxygen. The above solution was transferred into the autoclave under vacuum via a funnel tightly connected to the introduction valve of the autoclave. Then, the vessel was cooled in a liquid nitrogen bath, and ethylene (2 g, 71.4 mmol; 1% eq.) was introduced through the autoclave inlet valve while monitoring its amount by double weighing (i.e., measuring the difference of masses before and after filling the autoclave with ethylene). The autoclave was then allowed to warm to ambient temperature and heated to the target reaction temperature (60 °C) under mechanical stirring, and the evolutions of pressure and temperature were recorded. Suddenly, after ca. one hour of reaction at 60 °C, a fast high increase of pressure up to 70 bars occurred, immediately followed by a drop in pressure as low as 5 bars, constituting a complete reaction. The autoclave was then cooled in an ice bath for 30 min and no release of unreacted gas was noted following the opening of the vessel. The total product mixture (yellow solid) was recovered from the autoclave and characterized using ^19^F NMR. The evidence given by the absence of a signal at –70 ppm assigned to CF_2_CF_2_I end-group indicated the quantitative conversion of 5–iodo–3–oxaperfluoropentane–1–sulfonyl fluoride. The crude product was then dissolved in acetone and precipitated from water and finally dried under vacuum (20 × 10^−3^ bar, 50 °C) for 8 h, yielding 16.55 g of a yellowish powder (yield = 94%).

IR (cm^−1^) (Appendix A): 1460–1480 cm^−1^ and 1200–1220 cm^−1^: asymmetric and symmetric SO2 stretching vibrations; 1100–120 cm^−1^: C-F bond; 810–820 cm^−1^: S-F stretching vibration of the fluorosulfonyl group.

^1^H NMR (d-acetone, ppm, Appendix A): 3.18 (complex signal; ICH_2_CF_2_); 2.58 (complex triplet of triplets; CH_2_CF_2_).

^19^F NMR (d-acetone, ppm, Appendix A): +45 (SO_2_F); absence of signal at −70 ppm; −82.90 (complex signal; CH_2_CF_2_C***F_2_***O); −87.95 (dd; OC***F_2_***CF_2_SO_2_F); −113.05 (broad signal; CF_2_SO_2_F); −129.30 (CH_2_CF_2_; t, ^3^J_FF_ = 9.05 Hz).

Oxidation of the sulfonyl fluoride in I–CH_2_CH_2_CF_2_CF_2_OCF_2_CF_2_SO_2_F (5.23 g, 11.50 mmol) into SO_3_H was carried out first in presence of 0.89 g (12.04 mmol) of Li_2_CO_3_ in 20 mL of methanol at room temperature to lead to ICH_2_CH_2_CF_2_CF_2_OCF_2_CF_2_SO_3_Li (yield not determined). The latter was then acidified in presence of 15 mL of HCl (2N) to chemically change the –SO_3_Li end-group into sulfonic acid, thus producing 4.28 g of ICH_2_CH_2_CF_2_CF_2_OCF_2_CF_2_SO_3_H as a white powder, recrystallized in acetone. The overall yield from 1,1,2,2–tetrafluoro–2–(1,1,2,2–tetrafluoro–2–iodoethoxy)ethanesulfonyl fluoride was ca. 78%.

IR (cm^−1^) (Appendix A): 3000 and 1020 cm^−1^: S(O)2(OH) stretching vibrations of the sulfonic acid group.

^1^H NMR (d-acetone, ppm, Appendix A): 3.38 (complex signal; ICH_2_CF_2_); 2.83 (complex triplet of triplets; CH_2_CF_2_).

^19^F NMR (d-acetone, ppm, Appendix A): absence of signal at +45 (SO_2_F); −83.90 (CH_2_CF_2_C***F_2_***O); 89.60 (OC***F_2_***CF_2_SO_3_H); −119.20 (overlapping of CH_2_CF_2_ and CF_2_SO_3_H).

### 3.3. Methods of Characterization

The morphology and physical properties of the sample were studied through thermogravimetric analysis, differential scanning calorimetry, polarized optical microscopy, temperature-resolved small- (SAXS) and wide-angle X-ray scattering (WAXS), and computer simulation.

Thermogravimetric experiments were conducted using TGA Q500 from TA Instruments (New Castle, DE, USA) from 25 °C to 900 °C at a heating rate of 10 °C under a nitrogen flow of 100 mL/min.

Differential scanning calorimetry (DSC) measurements were carried out using a Polyma 214, Netzsch (Selb, Germany). Samples were placed in pierced aluminum pans and subjected to two heating−cooling cycles from 20 to 260 °C at a rate of 10 K·min^−1^.

Microscopic images of thin films were captured using a Carl Zeiss AxioScope A1 POL (Jena, Germany) optical microscope in polarized light with a 100× objective. The films, prepared from the melt between two cover glasses, were positioned on a Linkam LTS heating stage (Surrey, UK). A small amount of the sample was heated to 260 °C, above melting temperature, and then crystallized at two different cooling rates: 1 and 10 K·min^−1^. Images were captured with a 5 MP CMOS camera (ToupTek Photonics, Hangzhou, China).

Temperature-resolved SAXS and WAXS measurements were carried out at the ID02 beamline of the European Synchrotron Radiation Facility (ESRF) in Grenoble (France). The measurements were performed in transmission geometry with X-ray photons at an energy of 11.9 keV. The calibration of the norm of the reciprocal vector *s* ($\left|s\right|=\frac{2sin\theta }{\lambda }$ where *θ* is the Bragg angle and *λ* is the wavelength) was done using several diffraction orders of silver behenate (for SAXS) and corundum (for WAXS).

All-atom molecular dynamics (AAMD) simulations of I(CH_2_)_2_(CF_2_)_2_O(CF_2_)_2_SO_3_H melting under different temperature conditions were carried out as follows. The initial unit cell configuration was established through Monte Carlo annealing among a set of most common symmetry groups, with geometry optimization using the COMPASS force field [37]. The elementary configuration with the lowest energy was then replicated to form a periodic box measuring 100 Å × 30 Å × 30 Å, containing 288 I(CH2)_2_(CF_2_)_2_O(CF_2_)_2_SO_3_H molecules. The COMPASS force field was utilized for the AAMD simulations, and electrostatic interactions were computed using the PPPM method [38].

The system was equilibrated in the isobaric–isothermal ensemble (NPT) for 10 ns with a Berendsen barostat [39] and a Nosé–Hoover thermostat [40] at a temperature of *T* = 25 °C and a pressure of *P* = 1 atm. Subsequently, the simulation box was heated sequentially by 5 °C every 10 ns up to 280 °C. Statistics were gathered over the last 1 ns of every heating step across three independent heating procedures.

Radial distribution function (RDF) and static structure factor *S*(*k*) were utilized for characterizing the translational order of molecules within the box. Orientational ordering was monitored by calculating the first eigenvalue of nematic *Q*–tensor, $S=<\frac{3\;{cos}^{2}\gamma \;-\;1}{2}>$ [33], where *γ* is the angle between molecules’ first principal axis and the director. 

## 4. Conclusions

Initially, a model of the membrane was proposed based on the ethylenation of 1,1,2,2–tetrafluoro–2–(1,1,2,2–tetrafluoro–2–iodoethoxy) ethanesulfonyl fluoride followed by the chemical modification of SO_2_F group into SO_3_H function. The formation of the structure of ionomeric comb-like polymer precursors was carried out, and the dependencies of their self-organization on thermal history were established through both experimental and theoretical considerations. It was discovered that at room temperature, the compound exhibits a highly ordered crystal structure with a monoclinic lattice formed by almost-extended molecules. Heating to 150 °C results in partial disordering of fluorinated tails and a transition to a triclinic unit cell. During cooling from the isotropic state, the molecules form large domains of the smectic LC structure. This smectic phase acts as a pre-ordered state for crystallization into a layer-like monoclinic phase at room temperature. Experimental findings on the formation of the crystalline phase and crystal-to-crystal phase transitions were validated through computer modeling using molecular mechanics approaches. The emergence of a smectic phase for relatively short linear fluorinated molecules is significant for the synthesis of self-organizing comb-like ionomers with controlled morphology of ion-conductive channels. The formation of the smectic phase was confirmed through all-atom molecular dynamics simulations, and the temperatures of smectic–nematic and nematic–isotropic transitions upon heating were determined. It was demonstrated that theoretical results are in good agreement with experimental findings. The insights presented in this work can be utilized for the development of new types of fuel cells with enhanced control over their morphology and improved performance.

## Data Availability

Data are contained within the article and Appendix A.

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
