# Peer review of "Synthesis of Calamitic Fluorinated Mesogens with Complex Crystallization Behavior"

_molecules, 2023, doi:10.3390/molecules28248002_

Round 1

Reviewer 1 Report

Comments and Suggestions for Authors

In this manuscript, entitled “Synthesis of calamitic fluorinated mesogens with complex crystallization behaviour” authors synthesized calamitic fluorinated mesogens from ethylenation of 1,1,2,2-tetrafluoro-2-(1,1,2,2-tetrafluoro-2-iodoetoxy)ethanesulfonyl fluoride and studied the crystallization behavior at various temperature. The results are interesting, and authors used various techniques to study the crystallization behavior for the calamitic fluorinated mesogens. I recommend this manuscript for publication. I request authors to address the below comments while submitting the manuscript.   

1). Authors could provide a detailed synthetic scheme for the synthesis of ICH2CH2CF2CF2OCF2CF2SO3H.

2). Scheme-1 needs to be renamed as a synthesis of ICH2CH2CF2CF2OCF2CF2SO3H to provide the complete details.

3). Provide the complete characterization (1H, 13CNMR, HRMS, IR and Melting points) for all the intermediates and final compound.

4). The authors could provide the full spectra (1H and 13C) of Figure S4. 1H-NMR spectrum of ICH2CH2CF2CF2OCF2CF2SO3H

5). Authors need to provide a clear experimental procedure with mmol and equiv. of all the reactants and reaction conditions.   

6). How lithiation happens on I-CH2CH2CF2CF2OCF2CF2SO2F with K2CO3.

Comments on the Quality of English Language

The manuscript is clearly written and does not have major english corrections

Author Response

Dera Reviewer,

please find our oint by point aswers

 1). Authors could provide a detailed synthetic scheme for the synthesis of ICH2CH2CF2CF2OCF2CF2SO3H.

Answer : Scheme 1 in the main manuscript has been improved.

2). Scheme-1 needs to be renamed as a synthesis of ICH2CH2CF2CF2OCF2CF2SO3H to provide the complete details.

Answer : yes, this is done in the revised document.

3). Provide the complete characterization (1H, 13CNMR, HRMS, IR and Melting points) for all the intermediates and final compound.

Answer : We are sorry not to have been able to supply all the characterizations (e.g. 13CNMR, HRMS, and melting points) of all intermediates but the revised supporting information has been improved with additional .

4). The authors could provide the full spectra (1H and 13C) of Figure S4. 1H-NMR spectrum of ICH2CH2CF2CF2OCF2CF2SO3H

Answer: unfortunately, the 13C NMR spectrum could not be supplied since we donot have anymore sample.

5). Authors need to provide a clear experimental procedure with mmol and equiv. of all the reactants and reaction conditions.   

Answer : the experimental protocol has been extended in the revised manuscript, as requested.

6). How lithiation happens on I-CH2CH2CF2CF2OCF2CF2SO2F with K2CO3.

Answer: we are sorry, this has been a mistake: Li2CO3 was used to allow the formation of SO3Li has indeed occurred.

Sincerely,

Bruno Ameduri

Reviewer 2 Report

Comments and Suggestions for Authors

This manuscript by Anokhin, Ameduri and coworkers is an interesting contribution towards the understanding of self-organization of fluorinated precursors during PFSA membrane fabrication. The calamitic fluorinated mesogens were synthesized in a three-step procedure with good yields and the physical and morphology properties investigated by a range of techniques, including theoretical computer simulation.

Even though these procedures are well-described, it is surprising that full analytical data (1H, 13C, 19F, HRMS) was not reported for the intermediate and final synthetic products, only some spectra are shown in the supplementary information. The authors should describe these data in section 2 of the manuscript.

Furthermore, there are some typos to correct:

page 2, line 67: "understanding" instead of "understating"

page 2, line 78 and 83: "2-iodoethoxy" instead of "2-iodoetoxy"

page 2, line 81: "cyclohexyl" instead of "cycloheyl"

After these minor corrections, the manuscript should be accepted for publication.

Author Response

This manuscript by Anokhin, Ameduri and coworkers is an interesting contribution towards the understanding of self-organization of fluorinated precursors during PFSA membrane fabrication. The calamitic fluorinated mesogens were synthesized in a three-step procedure with good yields and the physical and morphology properties investigated by a range of techniques, including theoretical computer simulation.

Even though these procedures are well-described, it is surprising that full analytical data (1H, 13C, 19F, HRMS) was not reported for the intermediate and final synthetic products, only some spectra are shown in the supplementary information. The authors should describe these data in section 2 of the manuscript.

Furthermore, there are some typos to correct:

page 2, line 67: "understanding" instead of "understating"

Answer : this has been corrected in the revised version

page 2, line 78 and 83: "2-iodoethoxy" instead of "2-iodoetoxy"

Answer : this has also been corrected in the revised version

page 2, line 81: "cyclohexyl" instead of "cycloheyl"

Answer : this has also been corrected in the revised version

After these minor corrections, the manuscript should be accepted for publication.

We appreciate such comments

Sincerely

B. Ameduri